# A Rice R2R3-Type MYB Transcription Factor OsFLP Positively Regulates Drought Stress Response via OsNAC

**DOI:** 10.3390/ijms23115873

**Published:** 2022-05-24

**Authors:** Xiaoxiao Qu, Junjie Zou, Junxue Wang, Kezhen Yang, Xiaoqin Wang, Jie Le

**Affiliations:** 1Beijing Advanced Innovation Center for Tree Breeding by Molecular Design, Beijing University of Agriculture, Beijing 102206, China; quxiaoxiao721@sina.com; 2Key Laboratory of Plant Molecular Physiology, CAS Center for Excellence in Molecular Plant Sciences, Institute of Botany, Chinese Academy of Sciences, Beijing 100093, China; wangjunxues@163.com (J.W.); ykzdsp@163.com (K.Y.); 3Biotechnology Research Institute, Chinese Academy of Agricultural Sciences, Beijing 100081, China; zoujunjie@caas.cn

**Keywords:** OsFLP, OsNAC, rice, stomata closure, drought tolerance

## Abstract

Abiotic stresses adversely affect plant growth and the yield of crops worldwide. R2R3-MYB transcriptional factors have been found to be vital for plants to confer stress response. In *Arabidopsis*, FOUR LIPS (FLP, MYB124) and its paralogous MYB88 function redundantly regulated the symmetric division of guard mother cells (GMCs) and abiotic stress response. Here, OsFLP was identified as an R2R3-MYB transcriptional activator and localized in the nucleus. *OsFLP* was transiently induced by drought, salt stress and abscisic acid (ABA). Overexpression of *OsFLP* showed enhanced tolerance to drought and salt stresses. The stomatal density in *OsFLP-OE* plants was not changed, whereas the stomatal closure was sensitive to ABA treatment compared to wild-type plants. In contrast, *OsFLP*-RNAi plants had abnormal stomata and were sensitive to drought. Moreover, the transcripts of stomatal closure-related genes *DST* and *peroxidase 24 precursor*, which are identified as downstream of OsNAC1, were inhibited in *OsFLP*-RNAi plants. The yeast-one-hybrid assay indicated that OsFLP can specifically bind and positively regulate *OsNAC1* and *OsNAC6*. Meanwhile, stress response genes, such as *OsLEA3* and *OsDREB2A*, were up-regulated in *OsFLP*-OE plants. These findings suggested that OsFLP positively participates in drought stress, mainly through regulating regulators’ transcripts of *OsNAC1* and *OsNAC6*.

## 1. Introduction

Drought and salt stresses can significantly impair plants’ growth and development, leading to reduced crop productivity. The plant epidermal cell is the first barrier that separates a cell from the external environment and has a protective role as human skin. Stomata are the most important epidermal structure of terrestrial plants, and stomatal aperture and patterning control the water vapor exchange and ion transport. Therefore, stomatal pattern and movement play an important role in response to environmental stresses [1,2,3,4,5,6]. With the progression in studying stomatal development using the dicotyledonous model plant *Arabidopsis thaliana*, many genes related to the stomatal development pathway have been characterized, and some of them also participate in response to environmental stresses [3,4]. For example, stomatal density and distribution protein, SDD1, participates in drought and salt stresses [2]. Mitogen-activated protein kinases, MPK3 and MPK6, which are downstream of SDD1, can be activated by abiotic stresses [7,8,9,10]. Two R2R3-MYB transcriptional factors, FLP and its paralogous MYB88 act as main regulators in restricting the symmetrical cell division of the GMC, and also respond to drought and salt stresses [5,11,12]. The *flp-1 myb88* double mutants are more sensitive to drought and high-salt stresses [5]. MdMYB88 and MdMYB124 promote anthocyanin accumulation and H_2_O_2_ detoxification in response to cold stress in apples [13]. According to the ChIP-chip assay, FLP can directly target promoters of cell cycle genes, auxin−related genes, transcriptional factors and ubiquitin modification protein genes, indicating that versatile regulatory pathways may exist in FLP-mediated stress responses [11].

The NAC (NAM, AFAT and CUC) transcription factors are plant-specific gene families. NACs have been largely elucidated in various species, such as *Arabidopsis* [14,15], rice [16,17,18], soybean [19,20,21,22,23,24,25] wheat [26,27] and maize [28,29]. The NAC proteins play vital roles in diverse aspects of plant growth, development and stress responses [15,30,31,32,33,34]. In *Arabidopsis*, Tran and Fujita et al reported that three *NAC* genes, *ANAC019*, *ANAC055* and *ANAC072* were induced by drought, salinity, and/or low temperature, and overexpressing these genes showed enhanced stress tolerance [35,36]. A previous report revealed that there are 151 *NAC* genes predicted in rice [16]. The roles of the *NACs* in response to abiotic stresses have been extensively explored, and overexpression of some rice *NAC* genes can improve plants’ tolerance to abiotic stresses. For example, *SNAC1* can be specifically induced in the guard cells of rice under drought stress. *SNAC1*-overexpressing plants were more tolerant to drought and salt stresses through accelerating stomatal closing and increasing expression levels of a large number of stress-related genes [37]. Another, overexpression of rice *SNAC2* produced more grains and showed significantly increased tolerance to cold, salinity and dehydration stresses. The genomic expression profile of the *SNAC2*-overexpressing plants showed enhanced stress response genes [32,38]. It has also been reported that *OsNAC5* plays an important role in abiotic stress response, and OsNAC5 can directly regulate *OsCCR10* and contribute to drought tolerance by modulating lignin accumulation in roots [39,40,41,42]. Another stress-responsive gene, *OsNAC6*, which is a member of the ATAF subfamily, can be induced by abiotic and biotic stresses, such as cold, high salinity, drought, ABA, JA and blast disease. Overexpression of *OsNAC6* in rice resulted in enhanced tolerance to dehydration and high-salt stresses [33,43,44]. The overexpression of *OsNAC9* [45], *OsNAC10* [46], or *OsNAP* [47] improves drought and salinity tolerance in transgenic rice. Whereas, *OsNAC4* is a key positive regulator of plant hypersensitive cell death and its expression is negatively controlled by miR164 [31,48].

In this study, a single R2R3-MYB transcriptional factor, OsFLP, was characterized and its function in drought stress was analyzed in detail. The expression levels of *OsFLP* were induced by various abiotic stresses, including drought and salt. Overexpressing *OsFLP* significantly enhanced drought and salt tolerance, whereas reducing *OsFLP* expression levels not only led to abnormal GMC division and box-shaped stomata but also resulted in plants’ sensitivity to drought stress. Further study indicated that OsFLP can specifically bind to the promoter of *OsNAC1* and *OsNAC6* and positively regulate their expression levels. Our study provides new insights into enabling the development of high-yielding rice varieties with increased drought tolerance by using genetic engineering strategies.

## 2. Results

### 2.1. OsFLP Functions as a Transcriptional Activator and Is Localized in the Nucleus

AtFLP and its paralogous MYB88 contribute to stomatal development and abiotic stress in *Arabidopsis* [5,11]. There was only one FLP in rice, named OsFLP, which regulates the symmetrical division of rice stomatal development [49]. Multiple sequence alignment with homolog proteins of AtFLP, AtMYB88, AtMYB121, AtGL1 and OsMPS showed that R2 and R3 MYB DNA-binding domains present at the N-terminal region of OsFLP, and each domain has the H1, H2, H3 conserved motif, indicating that OsFLP is an R2R3-MYB protein (Figure 1A). Additionally, qRT-PCR analyses showed that the expression levels of OsFLP are higher in young shoots and mature leaves than those in the roots (Figure 1B). Transient expression of the OsFLP coding sequence fused with GFP in tobacco leaves revealed that obvious fluorescence signals are found in the nucleus (Figure 1C). To examine whether OsFLP has transactivation activity, the full-length cDNA of *OsFLP* was fused to the pGBKT7 vector with the GAL4 DNA-binding domain. The transcription factor OsNAC6, which has transactivation activity, was used as a positive control [44]. As shown in Figure 1D, only the transformants containing OsFLP or OsNAC6 exhibited obvious β-galactosidase activity after X-gal staining. Collectively, these results indicated that OsFLP is mainly localized in the nucleus and functions as a transcriptional activator.

### 2.2. Expression of OsFLP Is Induced by Abiotic Stresses

Previous report elucidated that AtFLP and its paralog AtMYB88 contribute to abiotic stress in *Arabidopsis* [5,11,12] (Appendix A). To confirm whether *OsFLP* is responsive to abiotic stresses, we investigated the expression patterns of *OsFLP* under different abiotic stresses by qRT-PCR. The results showed that *OsFLP* was induced by drought, salt and ABA treatment, but not by low temperature. Under drought treatment, the transcripts of *OsFLP* were significantly up-regulated at 2 h and reached a maximum level at 6 h, followed by a decrease at 12 h (Figure 2A). Meanwhile, transcript levels of *OsFLP* were increased by 200 mM of NaCl, showing a similar pattern to drought treatment (Figure 2B). Whereas *OsFLP* transcript levels were gradually increased for up to 12 h after ABA treatment (Figure 2C). However, the *OsFLP* transcripts were not significantly altered under the 4 °C treatment (Figure 2D). These results suggested that *OsFLP* was response to drought and salt stresses.

### 2.3. Overexpression of OsFLP in Rice Improves Plants Tolerance to Drought

To further investigate the function of OsFLP on drought resistance, the coding sequences of *OsFLP* driven by the *CaMV 35S* promoter were generated and transformed into Zhonghua 11. Of the 24 independent transgenic plants generated, 2 positive transgenic lines with high expression levels (about 28-fold in OE-#2 and 18-fold in OE-#3 compared to WT) were selected for further drought stress treatments (Figure 3D). Four-week rice seedlings grown under normal conditions were subjected to drought stress by withholding water (Figure 3A). After the withdrawal of water for 14 days and re-watered for 5 days, the leaves of the WT plants were wilted and showed an obvious dehydration phenotype, whereas *OsFLP*-overexpression plants grew better than WT plants (Figure 3B). Water loss assays using detached leaves showed that WT plants lose water significantly faster than the *OsFLP*-overexpression plants (Figure 3C). Consistent with this, the survival rate of WT under drought was only about 37%, whereas the survival rates of two *OsFLP*-overexpression lines were 63% and 51%, respectively (Figure 3E). Furthermore, *OsFLP*-overexpression lines also have a much higher fresh weight and plant height than the WT after drought stress (Figure 3F,G).

To further identify the role of OsFLP in salt stress, we first treated the *Arabidopsis flp−1 myb88* mutants with 50 mM, 100 mM and 125 mM of NaCl. Consistent with the previous report, *flp−1 myb88* mutants were more sensitive to salt stress (Appendix A). We then obtained the transgenic plants of *OsFLP* ectopic overexpression in *Arabidopsis* Col−0 background and examined their salt sensitivity (Appendix A). The results showed that there are no significant differences between Col-0 and the transgenic plants under normal growth conditions. However, after being treated with 150 mM of NaCl for two weeks, *OsFLP*−OE plants in the Col-0 background grew better and had a higher survival rate than Col−0 plants (Appendix A). Taken together, these data indicated that *OsFLP* positively contributes to the drought and salt tolerance.

### 2.4. Knockdown of OsFLP Decreased Drought Tolerance and Showed Non−Functional Stomata

To further explore the role of OsFLP in drought stress, RNAi transgenic lines targeting *OsFLP* were generated. Of the 57 independent transgenic plants generated, transcript levels of *OsFLP* in two lines, RNAi-#5 and RNAi-#23, were suppressed to 18% and 2%, respectively, in relation to the level in WT seedlings (Figure 4C). Further, an examination of plants’ sensitivity to drought treatment indicated that the two *OsFLP*-RNAi transgenic lines had lower survival rates than WT plants (Figure 4A,B). The survival rate was nearly 40% in WT, whereas was only ~20% or ~10% in the two *OsFLP*-RNAi lines (Figure 4C).

Deletion of *OsFLP* resulted in the misorientation of some GMC symmetric divisions and formed non-function stomata [49]. We also observed the no-pore box-shaped GCs (Guard Cells) and undivided GCs in *OsFLP*-RNAi lines (Figure 4D). There were about 14% to 27% abnormal stomata in the *OsFLP*-RNAi lines (Figure 4E). Whereas the total stomatal density was altered neither in *OsFLP*-RNAi nor in *OsFLP*-OE lines compared to WT plants (Figure 4F and Appendix A). Furthermore, we noticed that expression levels of *DST* and the *peroxidase 24 precursor*, which were related to stomatal movement, were dramatically suppressed in the *OsFLP*-RNAi transgenic plants (Figure 4G,H), suggesting that the drought sensitivity of *OsFLP*-RNAi transgenic might result from abnormal stomatal functions.

### 2.5. OsFLP Confers Plants Tolerance to Drought by Regulating Stomatal Closure

The stomatal pattern and movement control transpiration water loss of plants plays an important role in drought stress [2,28,37]. Loss of both *AtFLP* and *AtMYB88* function leads to the formation of stomatal clusters [12]. To investigate the response of stomata under drought stress, we measured stomatal closure of *Arabidopsis OsFLP*-OE and WT in detached leaves treatment with ABA treatment. In normal conditions, no significant differences in stomatal aperture were found between *OsFLP*-OE and WT plants. However, stomatal closure in the *OsFLP*−OE plants was more faster than that in WT, especially in the detached leaves treated with ABA for 1 h (Figure 5A,B). These results suggest that overexpression of *OsFLP* can promote stomatal closure.

To further identify the function of OsFLP in stomata, cross-species complementation tests were conducted. *pMUTE:OsFLP is* driven by *Arabidopsis MUTE* promoter (MUTE is a bHLH transcription factor and especially expressed in stomatal lineage) and *35S:OsFLP* were introduced into *Arabidopsis flp−1* mutant, respectively. The stomatal cluster (4-GC) was partially rescued from 29% in *flp−1* mutant to 16% in *35S: OsFLP-OE* lines, whereas it was completely rescued to normal in *pMUTE:OsFLP flp−1* plants (Appendix A), supporting that OsFLP is a conserved regulator in stomatal development.

### 2.6. OsFLP Regulated Diverse Stress Response Genes in Drought Stress through Targeting OsNAC1 and OsNAC6

The NAC family is known to possess diverse roles as transcription factors in plant development and especially in stress response [26,36,50,51]. Different NACs are also regulated by specific transcription factors, such as the MYB family [5]. To identify whether NAC transcription factors are the targets of OsFLP, OsNAC1 and OsNAC4, OsNAC5 and OsNAC6 were selected as potential candidates. The expression levels of *OsNAC1* and *OsNAC6* were significantly induced in *OsFLP*-OE plants and repressed in *OsFLP*-RNAi plants compared to WT. However, the expression levels of *OsNAC4* and *OsNAC5* were not altered in *OsFLP*-OE and *OsFLP*-RNAi plants (Figure 6A,B). These results indicated that *OsFLP* may positively regulate the transcriptional expression levels of *OsNAC1* and *OsNAC6*. To examine whether OsFLP directly binds to the promoter region of *OsNAC1* and *OsNAC6*, we first analyzed the MYB binding motif by using the JASPAR CORE database. The results showed that the promoters of *OsNAC1* and *OsNAC6* contain 11 and 23 MYB binding motifs, respectively (Appendix A). To directly confirm whether OsFLP binds to *OsNAC1* and *OsNAC6*, a yeast one-hybrid assay was conducted. The 2000 bp upstream of the *OsNAC1* and 2041 bp upstream of the *OsNAC6* transcriptional start sites were identified (Figure 6C), and the results implied that OsFLP directly binds to the promoters of *OsNAC1* and *OsNAC6*, but not to those of *OsNAC4* and *OsNAC5*.

Furthermore, the promoter sequences of *OsNAC1* and *OsNAC6* were truncated into several fragments. The results showed that OsFLP can bind to three fragments, −371 to −624 bp (−542 to −551: CAACCTAAGA; −614 to −623: TGTCCTAATT), −1377 to −1682 bp and −1682 to −2000 bp (−1769 to −1778: AACCCAACCA) upstream of the *OsNAC1* transcriptional start site, and bind to two fragments, −1 to −239 bp (−77 to −86:AACACTAGTA; −72 to −81: TATCCTACTA) and −239 bp to −561 bp (−291 to −300: TGACGTAAGC) upstream of the *OsNAC6* transcriptional start site (Figure 6D).

We also explored the expression patterns of *OsNAC1*, *OsNAC4*, *OsNAC5* and *OsNAC6* under salt treatment. Transcripts of *OsNAC1*, *OsNAC5* and *OsNAC6* were significantly increased in WT under salt treatment, whereas OsNAC1 and *OsNAC6* were not induced in *OsFLP*-RNAi plants (Appendix A–C). Moreover, drought response genes *OsLEA* and *OsDREB2A* were also repressed in *OsFLP*-RNAi plants (Appendix A).

Taken together, these results indicated that OsFLP plays a positive role in response to drought stress through directly binding to the promoter regions of *OsNAC1* and *OsNAC6*, which then regulates the expression levels of stress response genes.

## 3. Discussion

The MYB transcription factors are one of the largest plant transcription factor families and play pleiotropic roles in plant development and in various abiotic stresses [13,52,53,54]. Especially R2R3-type MYB transcription factors, which are the main subgroup in these processes. Thus far, FLP and its paralogue MYB88 are the only R2R3-MYB transcription factors, which have been reported, especially in regulating the plant stomatal terminal division in both *Arabidopsis* and rice [11,12,49], and also, FLP/MYBB sense and/or transduce salt and drought stress in the root by regulating stress response genes [5]. However, the biological functions of FLP/MYB88 in rice are still unclear. Therefore, the characterization of the function and mechanism of FLP/MYB88 in monocotyledon will contribute to the improvement of crop resistance. In this study, we characterized only a single *Arabidopsis* FLP/MYB88 homologous OsFLP in the rice genome, which is similar to rice OsCYCA2 and OsCDKB1 [55]. Multiple alignments revealed that OsFLP shared a high identity with AtMYB124, AtMYB88, AtMYB71 and OsMPS in the N-terminal and possessed the signature motifs of a typical R2R3-type MYB, indicating that OsFLP is a putative homologue of R2R3-type MYB. Results of the nuclear localization and transcriptional activity of OsFLP indicated that OsFLP is conserved with *Arabidopsis* FLP/MYB88.

The transcripts of *OsFLP* were induced by drought, salinity and ABA, indicating that OsFLP might be involved in abiotic stress. Therefore, *OsFLP* was ectopically expressed in rice and *Arabidopsis* to elucidate its function under drought and salinity stresses. The *OsFLP*-OE plants had greater tolerance to drought and salt treatment, displaying higher survival rates and lower leaf water loss rates than WT plants. Correspondingly, *OsFLP*-RNAi transgenic rice plants exhibited more sensitivity to drought stress and abnormal GCs (no-pore box-shaped GC and undivided GMC). However, we found that the stomatal density was affected neither in *OsFLP*-OE nor in *OsFLP*-RNAi transgenic lines. This indicated that the stress response may not result from the stomatal pattern, however, the *OsFLP*-RNAi transgenic rice plants’ sensitivity to drought may be partially caused by abnormal stomata. Recent work has demonstrated that OsFLP controls the orientation of GMC symmetric division [49]. Nevertheless, our study showed that *OsFLP*-RNAi transgenic plants mainly restrict the symmetric division of GMC, or displayed the correct symmetric division; however, they do not promote cell fate transition from GC-like cells in rice, and the cell wall thickens in the pore of GC-like cells, which are necessary for functional GCs. These discrepancies between our study and the previous study may be due to the deletion function of OsFLP plants used in the previous report. Here, the knockdown plants were used in this study. The low transcription levels of *OsFLP* in *OsFLP*−RNAi plants possibly rescues the asymmetric division.

Our results showed that the expression levels of stomatal closure-related genes *DST* and *peroxidase 24 precursor* in *OsFLP*-RNAi lines were lower than in WT plants. Furthermore, the ABA-induced stomatal closure assay indicated that the drought tolerance of *OsFLP*-OE plants was at least partially due to stomatal closure. Whereas, the *Arabidopsis* FLP/MYB88 participates in drought and salt stress responses via a mechanism likely to be independent of stomatal opening and closing [5]. These discrepancies may be due to functional differences between monocotyledon and dicotyledon. Just as *Arabidopsis* FLP/MYB88 regulate the frequency division of GMC [12], whereas OsFLP controls the orientation of the GMC division and the pore formation [49].

The NAC family protein plays an important role in stress response. Our results showed that only *OsNAC1* and *OsNAC6* were specially regulated by OsFLP, and OsFLP has no effect on regulating *OsNAC4* and *OsNAC5* transcripts regardless of their roles in stress response. Furthermore, Y1H analysis revealed multiple binding sites of OsFLP at the promoter of *OsNAC1* and *OsNAC6.* We further observed that *OsNAC1*, *OsNAC5* and *OsNAC6* were all induced under salt; even so, only *OsNAC1* and *OsNAC6* were not induced in *OsFLP*-RNAi plants, *OsNAC5* was still increased. Interestingly, different NAC has different regulated stress genes. For instance, none of the up-regulated genes in the *SNAC2*-overexpressing plants matched the genes up-regulated in the transgenic plants overexpressing other stress-responsive *NAC* genes reported previously [38]. These results suggest that OsFLP is involved in abiotic stress by regulating different *NACs* and broadly benefit and effectively improve the stress tolerance of rice.

In addition, we cannot exclude other R2R3-MYB transcription factors that regulate drought stress response through NACs, such as AtMYB60 (GC specific) and AtMYB61, which were characterized in *Arabidopsis* for their different roles in the regulation of stomatal closure. Additionally, NAC transcriptional factor may feedback control the expression of *OsFLP*, just as rice R2R3-MYB gene (*UGS5*) containing a core-binding sequence of putative NACRS in the promoter regions and its expression levels were also up-regulated in the *SNAC1*-overexpressing plants [37]. Further investigation of the relationship between OsFLP and other *NAC* genes is necessary for characterizing the regulatory mechanism under drought stress.

In summary, we indicated that *OsFLP* played a positive role in drought tolerance in rice (Figure 7). The transcript levels of *OsFLP* were regulated by drought, salinity and ABA. The overexpression of *OsFLP* lines has a higher survival rate and lower leaf water loss rate under drought treatment. Consistently, knockdown of *OsFLP* increased the drought sensitivity and displayed non-function stomata. Stomatal aperture bioassays showed that ectopic expression of *OsFLP* plants not only rescues the stomatal defects of the *Arabidopsis flp−1* mutant but also enhanced the stomatal closure, indicating that OsFLP is involved in drought stress partially by regulating the stomatal movement. The Y1H assay supported a preferential interaction between OsFLP and the promoter of *OsNAC1* and *OsNAC6*. These results indicate that OsFLP plays a crucial role in drought stress by regulating NAC transcriptional factors and their down-response genes. However, further study has to be carried out in order to know the exact functions of other components that interact with OsFLP to obtain a better understanding of the molecular mechanisms underlying OsFLP under abiotic stress.

## 4. Materials and Methods

### 4.1. Plant Materials and Growth Conditions

*Oryza sativa* L. *spp. japonica* cultivar Zhonghua 11 was used as a transformation recipient in the rice study. Rice seeds were soaked in water at 28 °C for 2 d, and then grown in a controlled growth chamber with 30 °C/22 °C day/night temperature cycles, 12 h/12 h light/dark illumination cycles, and 60–70% relative humidity. The Columbia−0 (Col−0) ecotype of *Arabidopsis thaliana* L. seeds was surface sterilized (40 s) in an aqueous solution of 30% (*w*/*v*) hydrogen peroxide and 85% (*v/v*) ethanol in a volume ratio of 1:4 (*v*/*v*), and then sowed on the surface of half-strength Murashige and Skoog (MS) medium supplemented with 0.8% agar and 1% sucrose. Plants were grown in a controlled temperature and photoperiod chamber at 22 ± 2 °C and 16 h/8 h light/dark illumination cycles.

### 4.2. Plasmid Construction and Generation of Transgenic Plants

To generate the construct of gene overexpression, the full-length cDNA of *OsFLP* was cloned into the pH7WG2D.1 vector by using gateway technology and LR Clonase TM II Enzyme Mix (Invitrogen). The recombinant plasmids were confirmed by sequencing before the transformation into Zhonghua 11 and Col-0. To generate RNAi transgenic plants against *OsFLP*, the conserved sequences from base pair (bp) 78 to 310 of *OsFLP* cDNA were amplified and cloned into the pTCK303 vector. The primer sequences used in this study are listed in Appendix A.

Transgenic plants were obtained as previously described [55]. We selected the T1 seeds on 1/2 MS liquid medium supplemented with 50 μg/mL of hygromycin. The positive transgenic lines were transplanted to soil and harvested T2 seeds by the individual plant. These seeds were further selected with hygromycin, and the ratio of nearly 3:1 positive and WT plants was selected and planted by the individual. T3 transgenic lines with a positive rate of 100% were selected with hygromycin as homozygous lines for a subsequent experiment.

### 4.3. Real-Time Quantitative PCR Analysis

The qRT-PCRs were performed by using SYBR Premix Ex Taq™ (TaKaRa) with a Corbett RG3000 (Corbett Research, Mortlake, Australia). The *OsACTIN2* gene was used as an internal control. The primer sequences are listed in Appendix A. All qRT-PCRs were carried out with independent biological replicates and three technical repeats.

### 4.4. Subcellular Localization and Fluorescence Observation

The full-length *OsFLP* CDS fragment was cloned into the pCAMBIA1300 vector resulting in the *35S:OsFLP-GFP* plasmid. This construct was transformed into *Agrobacterium tumefaciens* GV3101. The *Agrobacteria* with GFP fusion plasmid and *35S:H2B-mCherry* were suspended in an activation buffer (10 mM MES, 50 mM MgCl**_2_**, 100 mM AcetoSyringone), and were co-infiltrated into leaves of *N. benthamiana*. After dark growth for 36 h, the GFP signal was observed using an Olympus FV1000MPE multiphoton laser scanning microscope, with a 488 nm wavelength for the GFP signal and a 552 nm wavelength for the mCherry signal.

### 4.5. Transactivation Experiments in Yeast

The activation domain of OsFLP was examined by a yeast assay system. The full-length OsFLP CDS was cloned into the DNA-binding domain vector pGBKT7 (Clontech, Palo Alto, CA, USA). GAL4 DNA binding domain-OsFLP fusion protein would be produced when the yeast cells AH109 were transformed. The transformed yeast was grown on SD plates without tryptophan, Histidine and adenine. The plates were incubated for 3 days and then subjected to a β-galactosidase assay. The vector pGBKT7 was expressed as a negative control and OsNAC6 as a positive control.

### 4.6. Leaf Water Loss Assay

*Arabidopsis* and rice leaves of 4-week-old plants were detached, weighed immediately on a piece of weighting paper, and then place on the laboratory bench. The fresh weight (FW) was measured at the time intervals indicated at room temperature (22 °C). Water loss was calculated from the decrease in FW compared with time zero.

### 4.7. Assessment of Drought and Salt Tolerance

For drought stress tolerance testing, transgenic rice seeds were germinated on water containing 50 mg/l of hygromycin for a week and the seedlings were transferred into soil. The seedlings were grown for another three weeks and were withheld from water until the leaves displayed wilted and drought-stressed phenotypes (the degree of leaf-rolling), and were then re-watered for 5 days and photographed. For *Arabidopsis* drought stress, seedlings were grown on 1/2 medium for 7 d and then transferred to soil in the growth chamber with 16 h light/ 8 h dark. Seedlings grown for two weeks were not irrigated for 15 d and re-watered for 3 days. For salt treatment, 5-day-old *Arabidopsis* seedlings were transferred to a 1/2 MS medium containing different concentrations of NaCl and grown in a vertical orientation until the leaves turned to albino.

### 4.8. Stomatal Aperture Bioassays

The stomatal aperture was analyzed in abaxial epidermal cells from true leaves of *Arabidopsis*. Three weeks of rosette leaves were floated in buffer solution (50 mM KCl, 0.1 mM GaC**l**_2_ and 10 mM Mes-KOH, pH 6.1) for 3 h for the stomata full opening. Subsequently, the leaves were transferred to a buffer solution containing 10 µM of ABA for 1 h and 2 h. Then, the epidermis was separated and placed onto a slide and the images were photographed. The aperture width of each stomatal pore was determined by the image.

### 4.9. Yeast One-Hybrid Assay

The full-length cDNA sequence of *OsFLP* was amplified by using the primers listed in Appendix A and cloned into pGADT7 vectors (Clontech). The promoter of *OsNAC1*, *OsNAC4*, *OsNAC5* and *OsNAC6* were fused with the LacZ reporter vector. The 2000 bp promoter fragment of *OsNAC1* was truncated with 7 fragments (−2000 bp to −1682 bp, −1682 bp to −1377 bp, −1377 bp to −980 bp, −980 bp to −624 bp, −624 bp to −371 bp, −371 bp to −216 bp, −216 bp to −1 bp), and fused with the *LacZ* reporter vector, respectively. The 2041 bp promoter fragment of *OsNAC6* was truncated with 3 fragments (−2041 bp to −561 bp, −561 bp to −239 bp, −239 bp to −1 bp), and fused with the *LacZ* reporter vector, respectively. Afterward, the fused *OsFLP*-pGADT7 and fragments fused LacZ reporter vectors were co-transformed into the EGY48 yeast strain and performed X-gal staining.

### 4.10. Statistical Analysis

All experiments were carried out with three biological repeats and qRT-PCR also had three technical repeats. The student’s *t*-test was used to analyze the significant differences with the statistics *t*-test of SigmaPlot 10.0 software. We took *p* < 0.05 and *p* < 0.01 to indicate the significant differences.

## 5. Conclusions

In this study, we cloned and characterized a rice R2R3-MYB transcription factor gene, OsFLP, which is localized in the nucleus and possesses transactivation activity. OsFLP was induced by diverse abiotic stresses and *OsFLP*-overexpressing rice plants significantly enhanced drought resistance and salinity tolerance in rice and *Arabidopsis*. Our findings demonstrated that OsFLP acts as a positive regulator to mediate plant drought tolerance by regulating stomatal closure and expression of stress response genes. OsFLP can directly bind to the promoter sequences of *OsNAC1* and *OsNAC6*, especially. Our findings expanded the understanding of complex abiotic stress signal networks, which might show great promise for the genetic improvement of stress tolerance in rice.

## 6. Patents

A patent for this invention has been applied for. The Patent no. is ZL 2014 1 0072652.6. The inventors are listed as follows: Jie Le; Xiaoxiao.Qu; Junjie Zou.

## Figures and Tables

**Figure 1 ijms-23-05873-f001:**
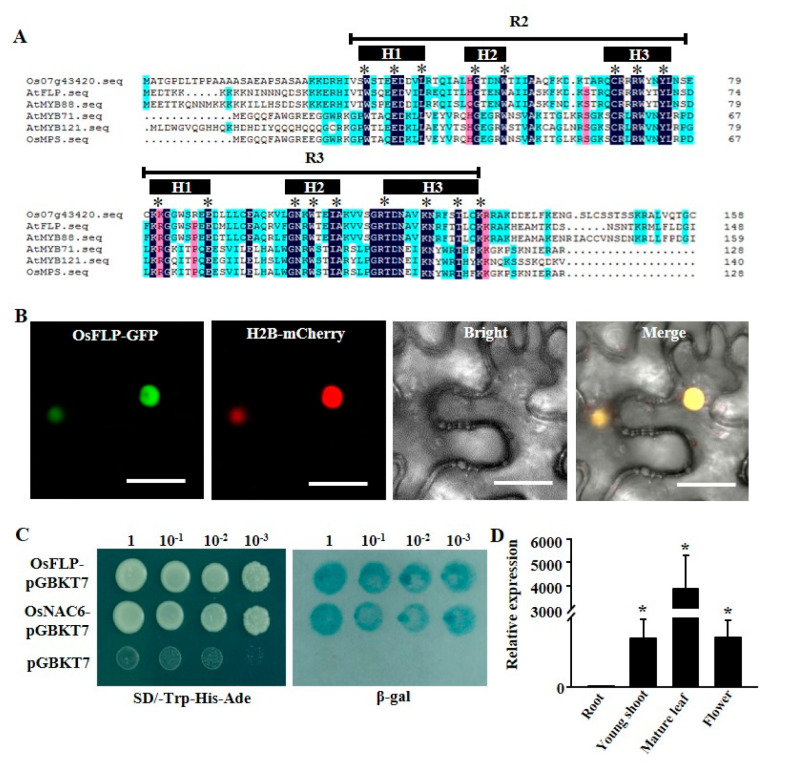
Category, subcellular localization, transcriptional activation and tissue expression of OsFLP. (**A**) Alignment of amino acid sequences of the OsFLP (Os07g43420) DNA-binding domain and homologous R2R3−-MYB proteins in A. thaliana (AtFLP/At1g14350, AtMYB88/At2g02820, AtMYB71/At3g24310, AtMYB121/At3g30210) and O. sativa (OsMPS/Os02g40530). (**B**) The subcellular localization of OsFLP in leaf epidermis cells of tobacco. OsFLP was fused with GFP. H2B-mcherry was the nucleus localization marker. OsFLP is mainly localized at the nucleus. Bars = 20 µm. (**C**) Analysis of transcription activation of OsFLP. The fusion protein of the GAL4 DNA-binding domain and OsFLP were expressed in yeast strain AH109. The vector pGBKT7 was expressed as a negative control and OsNAC6 as a positive control. The transformed yeasts were grown on SD plates without tryptophan, histidine and adenine. The plates were incubated for 3 days and then subjected to a β-galactosidase assay. (**D**) Quantitative real−time PCR (qRT−PCR) was performed to analyze the expression of *OsFLP* in different tissues, including roots, young shoots, mature leaves and flowers. High expression was detected in young shoots, mature leaves and flowers. Data represent means ± SD (*n* = 3 replicates). The statistically significant differences are indicated with asterisks: * *p* < 0.05.

**Figure 2 ijms-23-05873-f002:**
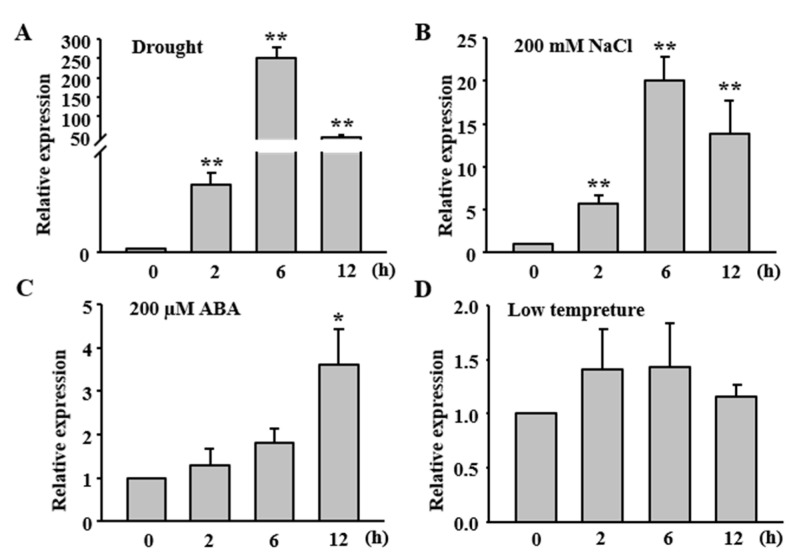
*OsFLP* transcripts were induced by drought, salt and ABA treatments. (**A**) qRT−PCR analysis of *OsFLP* under drought stress (20% PEG), (**B**) salt stress (200 mM NaCl), (**C**) ABA (200 µM) and (**D**) cold stress (4 °C). Two−week rice seedlings were treated for 2 h, 6 h and 12 h. Data represent means ± SD. Asterisks indicate a significant difference to control (Student’s *t*−test, ** *p* < 0.01, * *p* < 0.05).

**Figure 3 ijms-23-05873-f003:**
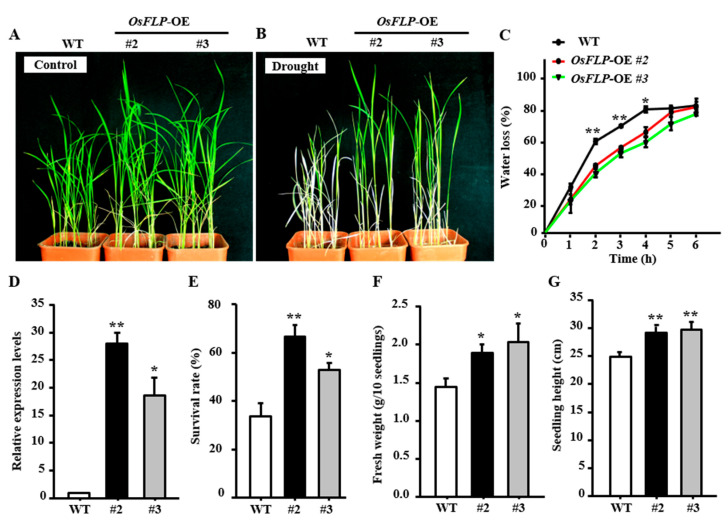
Overexpressing *OsFLP* enhanced plants’ drought tolerance. (**A**) Four-week rice seedlings grown under normal conditions. (**B**) Four-week rice seedlings were not watered for 14 days and re-watered for 5 days. (**C**) Water loss of detached leaves. Error bar means ± SD. (**D**) The expression levels of *OsFLP* transgenic lines were detected by qRT-PCR. Error bar means ± SD. (**E**) Analysis of survival rate after drought treatment. (**F**) Fresh weight of wild-type and transgenic lines after drought treatment. (**G**) Seedling height of wild-type and transgenic lines after drought treatment. Error bar of (**E**−**G**) means ± SE (*n* > 36 plants). Asterisks indicate a significant difference between *OsFLP*−OE lines to wild-type. (Student’s *t*-test, ** *p* < 0.01, * *p* < 0.05).

**Figure 4 ijms-23-05873-f004:**
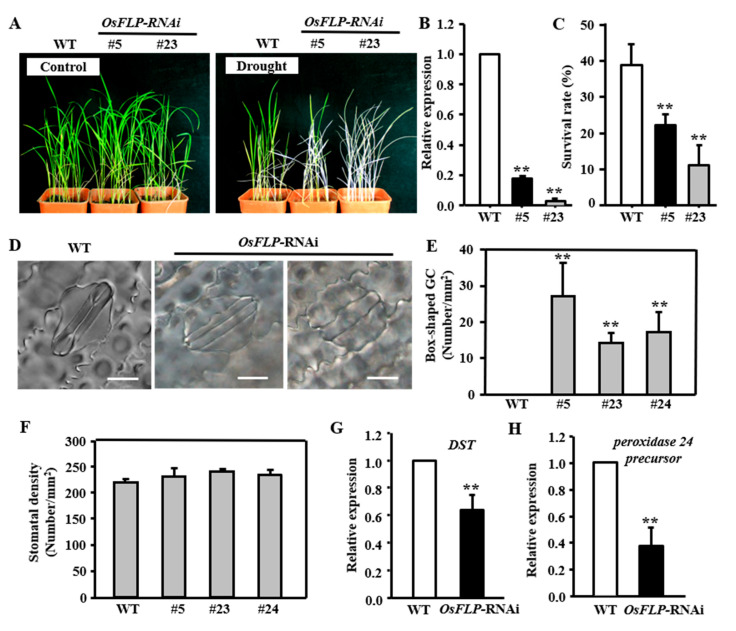
Knockdown of *OsFLP* results in drought sensitivity and abnormal stomata. (**A**) Four-week rice seedlings were grown under normal conditions and were not watered for 10 days and re-watered for 5 days. (**B**) The expression levels of transgenic lines were detected by qRT-PCR. Error bar means ± SD. (**C**) Analysis of survival rate after drought treatment. Error bar means ± SE (*n* > 36 plants). (**D**) Abnormal box-shaped guard cells were observed in *OsFLP*-RNAi lines. Bars = 10 µm. (**E**) The number of box-shaped guard cells was increased in three *OsFLP*-RNAi lines. Error bar means ± SD. (**F**) No significant difference in stomatal density was found between *OsFLP*-RNAi and WT mature leaves (*n* = 10 plants). (**G**,**H**) The transcript level of stomatal closure-related genes *DST* and *peroxidase 24 precursor* were significantly suppressed in *OsFLP*-RNAi lines. Error bar means ± SD. Asterisks indicate a significant difference to wild-type (Student’s *t*-test, ** *p* < 0.01).

**Figure 5 ijms-23-05873-f005:**
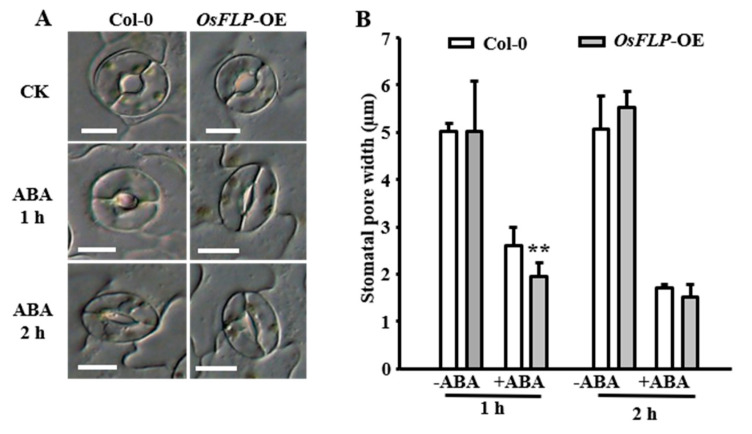
Overexpression of *OsFLP* enhanced the stomatal closure. (**A**) Representative images of Col-0 and *OsFLP*-OE stomata showing stomatal closure in response to ABA treatment for 1 h and 2 h. Bars = 10 µm. (**B**) Quantification of stomatal pore width in Col−0 and *OsFLP*-OE plants in response to 10 μM exogenous ABA; the stomatal aperture is indicated by the ratio of stomatal width. Data presented are the mean ± SE of three independent biological replicates. Significant differences in comparison with the WT are indicated as ** *p* < 0.01 (Student’s *t*-test).

**Figure 6 ijms-23-05873-f006:**
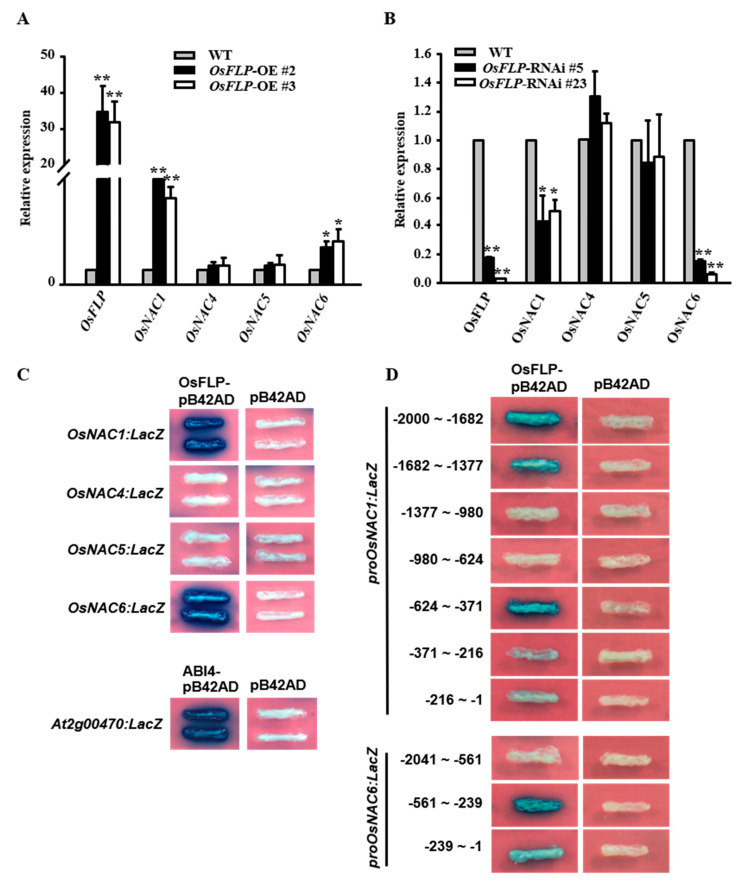
OsFLP positively regulates the *NAC* genes and specially binds to the promoters of *OsNAC1* and *OsNAC66*, respectively. (**A**) Expression levels of *OsFLP*, *OsNAC1* and *OsNAC6* were increased in *OsFLP*-overexpression lines. Error bar means ± SD. (**B**) Expression levels of *OsFLP*, *OsNAC1* and *OsNAC6* were reduced in *OsFLP* knockdown lines. Error bar means ± SD. (**C**) *LacZ* reporter genes were strongly activated and driven by *OsNAC1* and *OsNAC6* promoters in yeast. *LacZ* reporter driven by *At2g00470* was used as a positive control. (**D**) The full length of the promoter region of *OsNAC1* was truncated into seven fragments. Three fragments of −645 bp to −371 bp, −1682 bp to −1377 bp and −2000 bp to −1682 bp were identified as the binding regions of OsFLP. The full length of the promoter region of *OsNAC6* was truncated into three fragments. Two fragments of −561 bp to −239 bp and −239 bp to −1 bp were identified as the binding region of OsFLP. Asterisks indicate a significant difference to wild-type (Student’s *t*−test, ** *p* < 0.01, * *p* < 0.05).

**Figure 7 ijms-23-05873-f007:**
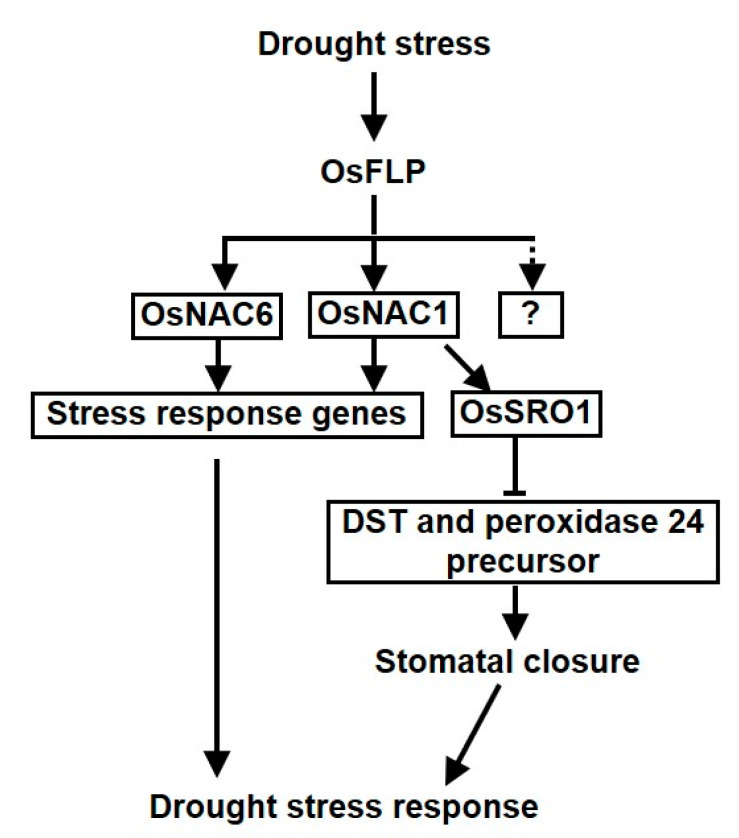
Proposed working model of OsFLP mediated drought stress response in rice. Under drought stress conditions, *OsFLP* expression levels are highly induced. Then, OsFLP specifically binds to the promoter region of *OsNAC1* and *OsNAC6* to increase the transcriptional expression of these genes. On the one hand, OsNAC1 and OsNAC6 can exclusively induce stress response genes; and on the other hand, *OsNAC1* enhanced the stomatal closure by inhibiting the *DST* and *peroxidase 24 precursor*. Ultimately, rice plants confer drought tolerance.

## Data Availability

Not applicable.

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
