# Peer review of "A Rice R2R3-Type MYB Transcription Factor OsFLP Positively Regulates Drought Stress Response via OsNAC"

_ijms, 2022, doi:10.3390/ijms23115873_

Round 1

Reviewer 1 Report

The present study focuses on investigating the functional roles of a rice R2R3-type MYB TF, OsFLP. The authors found that OsFLP gene is inducible in drought, salinity stress, and ABA treatment in rice. OsFLPox lines exhibited a significant increase in resistance to drought stress whereas OsFLP knockdown lines showed a significant decrease. The stomata assays showed the stomata closure, but no density, was affected concerning the drought stress phenotype of OsFLP. Finally, the authors showed that OsFLP interacts with the promoters of OsNAC1 and OsNAC6 in yeast, of which expression was affected by the overexpression and knockdown of OsFLP. In my opinion, this study has an interesting story that can be considered for publication in this journal. The results could be a good starting point for further downstream analyses and thus useful for the research community. However, the authors should address the following comments.

Major issues:
1) The title needs to be modified because the authors checked the interaction of OsFLP with OsNACs only in yeast. There are no in vivo (e.g., ChIP) and in vitro (e.g., EMSA) data in this study. I would suggest removing “direct targeting” to avoid any misleading. 
2) Method section needs to be substantially improved to provide sufficient information for readers. In addition, there is no method for Fig. 1 (transactivation assay and confocal imaging).
3) Fig. 1c: a nuclear maker (e.g., DAPI) needs to be shown to make sure that GFP is overlapped with the nuclear signal. 
4) provide information on whether the transgenic lines (both overexpression and knockdown lines) are homozygous for the transgene gene and knockdown allele, respectively.  
5) Figure legends need to be improved. For example, in Fig. 1B, how many biological replicates were used? What are bars (SD or SE)? 

Minor issues:
There are many English issues. I am pointing out a few as follows, but the authors need to make sure that the text is in a good shape: 
Line 36 – Arabidopsis thaliana needs to be italic
Line 120 – response > responsive
Line 155 – The expression levels of “OsFLP”

Reviewer 2 Report

  1. The authors used the way of gene overexpression to prove the function of rice OsFLP under drought stress and explored the signaling network, in my opinion, the scientific part looks reliable.
  2. The authors need to check the scientific names of plants and the names of genes to make sure they are italic throughout the manuscript.
  3. Avoid using first-person writing throughout the manuscript, e.g. we, our, etc.
  4. The text layout has some errors, it needs to be checked and corrected.
  5. L12: yield of crops - the yield of crops
  6. L13: have been found to be vital for plant - have been found to be vital for plants
  7. L25: that OsFLP positively participate - that OsFLP positively participates
  8. L32: that separate a cell from external environment and has a protection role on the human skin - that separates a cell from the external environment and has a protection role on human skin
  9. L125: similar with - similar to
  10. L149: rice seedling - rice seedlings
  11. L230: such as MYB family - such as the MYB family
  12. L232: as potential candidate - as potential candidates
  13. L240: bind - binds
  14. It has a conclusion in L422-435 and another conclusion is in L494-502, they look similar and need to be combined.
  15. A section on statistics should be added to M&M.
  16. Significant differences should be applied to figure 1B.
